# Quantifying Liver Heterogeneity via R2*-MRI with Super-Paramagnetic Iron Oxide Nanoparticles (SPION) to Characterize Liver Function and Tumor

**DOI:** 10.3390/cancers14215269

**Published:** 2022-10-27

**Authors:** Danny Lee, Jason Sohn, Alexander Kirichenko

**Affiliations:** 1Radiation Oncology, Allegheny Health Network, Pittsburgh, PA 15012, USA; 2Radiologic Sciences, Drexel University College of Medicine, Philadelphia, PA 19104, USA

**Keywords:** super-paramagnetic iron oxide nanoparticle, SPION, liver parenchyma, T2*-MRI, R2*-MRI, quantifying liver heterogeneity, auto-contouring, resecting liver surgery, liver radiation treatment planning, hepatic Kupffer cells

## Abstract

**Simple Summary:**

Super-paramagnetic iron oxide nanoparticles (SPIONs) are phagocytized by the hepatic Kupffer cells (KC) in the liver and shorten MRI signals within the volume of functional liver parenchyma (FLP) where KCs are found. However, malignant tumors lacking KCs exhibit minimal signal change, resulting in increasing liver heterogeneity. This study investigates whether SPIONs improve liver heterogeneity on R2*-MRI to characterize FLP and non-FLP (i.e., tumor, hepatic vessels, liver fibrosis and scarring associated with hepatic cirrhosis, prior liver-directed therapies or hepatic resection). By using SPIONs, liver heterogeneity was improved across two MRI sessions with and without an intravenous SPION injection, and the volume of FLP was identified in our auto-contouring tool. This is a desirable technique for achieving more accurate characterizations of liver function and tumors during radiation treatment planning.

**Abstract:**

The use of super-paramagnetic iron oxide nanoparticles (SPIONs) as an MRI contrast agent (SPION-CA) can safely label hepatic macrophages and be localized within hepatic parenchyma for T2*- and R2*-MRI of the liver. To date, no study has utilized the R2*-MRI with SPIONs for quantifying liver heterogeneity to characterize functional liver parenchyma (FLP) and hepatic tumors. This study investigates whether SPIONs enhance liver heterogeneity for an auto-contouring tool to identify the voxel-wise functional liver parenchyma volume (FLPV). This was the first study to directly evaluate the impact of SPIONs on the FLPV in R2*-MRI for 12 liver cancer patients. By using SPIONs, liver heterogeneity was improved across pre- and post-SPION MRI sessions. On average, 60% of the liver [range 40–78%] was identified as the FLPV in our auto-contouring tool with a pre-determined threshold of the mean R2* of the tumor and liver. This method performed well in 10 out of 12 liver cancer patients; the remaining 2 needed a longer echo time. These results demonstrate that our contouring tool with SPIONs can facilitate the heterogeneous R2* of the liver to automatically characterize FLP. This is a desirable technique for achieving more accurate FLPV contouring during liver radiation treatment planning.

## 1. Introduction

Kupffer cells (KC) are resident hepatic macrophages [1,2,3] and constitute 80–90% of the tissue macrophages present in the whole body [4]. KCs, as one of the major supporting cells, are responsible for maintaining liver function and preventing disease [4]. Therefore, magnetic resonance imaging (MRI) that targets resident liver macrophages [5,6] is often used to identify the volume of functional liver parenchyma (FLPV) using MRI contrast agents [7,8].

The use of a super-paramagnetic iron oxide nanoparticle (SPION)-based contrast agent (SPION-CA) [7,8] is gaining favor as an alternative to gadolinium-based contrast agents [9] to view primary or metastatic liver cancers since it is safer for patients with renal insufficiency [10]. SPIONs are trapped by KCs within the FLPV and shorten MRI signals. However, malignant tumors of hepatocellular carcinoma (HCC) and metastasis lacking KCs both exhibit minimal signal change. This results in increased liver heterogeneity to characterize between FLP and non-FLP (i.e., tumor, hepatic vessels, liver fibrosis and scarring associated with hepatic cirrhosis) [9]. After a single injection, SPIONs remain within KCs for several weeks, which allows for accurate tumor delineation and the subsequent avoidance of the FLPV during three-dimensional conformal radiation treatment planning [11,12,13].

T2*-MRI, the effective transverse relaxation time of T2 [14], is often used as the primary non-invasive approach to quantify the magnetic field inhomogeneity of iron deposition in liver [15,16]. Furthermore, by using T2*-MRI with a SPION-CA, malignant liver regions become more conspicuous due to susceptibility-related signal loss from SPION uptake with respect to the functional background liver [17,18]. Similarly, SPION-CAs safely label macrophages for MRI-based cell-tracking, leading to a considerable drop in T2* in the liver [19]. When T2* is transformed into reciprocal R2* (i.e., R2* [Hz] = 1000/T2* [ms]), it shows a strong correlation with hepatic iron deposition in the liver [20] as well as the breath-hold gradient echo (GRE)-based 2D and 3D T2*- and R2*-MRI. For these reasons, T2* is increasingly used to image the entire liver in a shorter acquisition time [21].

We previously demonstrated that functional radiation treatment planning for liver stereotactic body radiotherapy (SBRT) with a 99mTc sulfur colloid SPECT/CT allows the identification and avoidance of Kupffer-cell-rich FLPVs in patients with hepatocellular carcinomas and advanced hepatic cirrhosis by lowering toxicity and promoting greater local tumor control [22,23,24,25,26]. Our current study utilized T2*- and R2*-MRI to measure the susceptibility-related signal changes enhanced by SPION uptake in the hepatic Kupffer cells. There is currently no study which utilizes the change in R2* relaxation rates after SPION injection for quantifying liver heterogeneity to characterize FLP and non-FLP. Hence, our current work tests if SPIONs that enhance liver heterogeneity in the R2* of the liver can be utilized to characterize liver function and tumors while also developing an auto-contouring tool to determine the FLPV. The liver image is overlaid with the FLPV to evaluate the efficiency of the auto-contouring tool in the hope of protecting healthy tissue during treatment.

## 2. Materials and Methods

In this IRB-approved study, patients with primary and metastatic liver cancers underwent two MRI sessions to acquire T2* image sets before and after the IV injection of a SPION-CA. Voxel-wise R2* relaxation rates in the livers were calculated, and the voxel-wise R2* was analyzed as a function of SPION-enhanced liver heterogeneity to characterize FLP and non-FLP.

### 2.1. Study Design

This study was comprised of six steps (Figure 1). First, T2* image sets were acquired before and after the IV SPION-CA injection (Figure 1a) using a 1.5T Elekta Unity MR-Linac (Elekta; Stockholm, Sweden). The two sets of pre- and post-SPION T2* image sets were transferred to MiM software (v7.0.6, MIM Software Inc, Cleveland, OH, USA). The liver and tumor contours, manually delineated by a physician for radiation treatment planning, were then copied to both pre- and post-SPION T2* image sets (Figure 1b) in the MiM. Then, both liver contours of the pre- and post-SPION T2* image sets were extracted by binary masking for the voxel-wise 3D liver volumes (Figure 1c). Next, we calculated R2* relaxation rates [27] for R2* liver maps (Figure 1d) and auto-contoured FLPV (see Figure 1e). Finally, the FLPV was overlaid on the liver image. This allowed us to evaluate the efficiency of an in-house auto-contouring tool (Figure 1f) to improve conformal avoidance for further uses of FLPV during radiation treatment planning.

### 2.2. Patients

Twelve patients (aged 53–86 years) with primary (*n* = 8) and metastatic liver tumors (*n* = 4) were enrolled in an Intuitional Review Board (IRB)-approved study at Allegheny Health Network Cancer Institute (AHN-CI), Pittsburgh, PA, USA as part of a registered clinical trial study (NCT04682847). All patients underwent two MRI sessions using a 1.5T Philips MR scanner hybrid with Elekta Unity MR-Linac (Elekta; Stockholm, Sweden) before and after the IV injection of the SPION-CA Ferumoxytol^®^ (Feraheme, AMAG Pharmaceuticals, Waltham, MA, USA).

### 2.3. Two Pre- and Post-SPION MRI Sessions

The first MRI session (pre-SPION) was performed to acquire baseline image sets and allow liver patients to become familiar with MRI. For liver imaging, a 2D T2*-MRI with 16 echoes and a 3D T2*-MRI with 3 echoes sequences were used in conjunction with an exhalation breath-hold (Ex-BH) for motion management.

In a 2D turbo field echo (TFE) T2*-MRI, the imaging parameters were repetition time (TR)/16 echoes (TEs) = 41.8 ms/every 2.4 ms in between 2.3 ms and 39.5 ms, field of view (FOV) = 397 × 397 mm^2^, pixel size = 1.55 × 1.55 mm^2^, image matrix = 256 × 256, thickness = 5 mm, flip angle = 15° and bandwidth = 478 Hz. The Ex-BH was verbally instructed by an attending therapist, (i.e., breathe-in, breathe-out, breathe-in, breathe-out, and hold your breath), and each image set took 10 to 12 s; 81 images in total were acquired.

In a 3D TFE T2*-MRI, the imaging parameters were repetition time (TR)/three echoes (TEs) = 16.9 ms/4.6 ms, 9.2 ms and 13.8 ms, FOV = 400 × 400 mm^2^, pixel size = 1.18 × 1.18 mm^2^, image matrix = 352 × 352, thickness = 2.5 mm, flip angle = 13° and bandwidth = 249 Hz. An Ex-BH was verbally instructed by an attending therapist, and each image set took 20 to 24 s; 137 images in total were acquired.

After the first MRI session, 2 mg/kg Ferumoxytol^®^ was administered by IV infusion for a minimum time of 15 to 20 min while a nurse monitored the patient. The second MRI session with the same MRI sequences was repeated either 48 or 72 h after the Ferumoxytol^®^ infusion so that SPIONs trapped within hepatic KCs and altered relaxation times of T2* in residual liver parenchyma would improve heterogeneity in the liver. All pre- and post-SPION T2* image sets acquired during the first and second MRI sessions were transferred to MiM in a Digital Imaging and Communications in Medicine (DICOM) format.

### 2.4. Calculating R2* Relaxation Rates of Liver

This study calculated the R2* relaxation rates of pre- and post-SPION T2* image sets to characterize the heterogeneity of R2* in the liver. Firstly, the voxel values (i.e., a 16-bit grayscale in between 0 and 65,536) of the liver (see Figure 1c) were used to calculate T2* in the liver (see Figure 1d) using the Equation (1) [27].
S(t) = S_0_ × e^−t/T2*^, (1)
where t = TE, S = measured data, S_0_ = multiplicative constant and T2* = effective transverse relaxation time.

The voxel-wise T2* relaxation time of the liver was initially calculated per echo. Thus, 16 and 3 T2* echo data sets were prepared using 2D T2* with 16 echoes and 3D T2* with 3 echoes, respectively.

Finally, voxel-wise R2* relaxation rates of the liver as an R2* liver map were calculated using the Equation (2).
R2* (Hz) = 1/T2* (ms).(2)

In the middle of the study, it was necessary to perform a 2D T2*-MRI for four liver cancer patients so we could calculate 2D R2* in the liver and analyze the tendency of R2* to change along 16 echoes. The 3D T2*-MRIs were performed for 12 liver cancer patients with primary or metastatic liver cancer to calculate the 3D R2* in the liver across three echoes and further characterize the heterogeneity of the R2* liver map with SPION uptake in the liver.

### 2.5. Characterizing Liver Heterogeneity

This study utilized the heterogeneity of R2* liver maps to characterize FLP and non-FLP through two approaches for auto-contouring voxel-wise FLPV in the following manner (Figure 2):The voxel values of the R2* liver map in tumor and liver contours were extracted in two approaches (see Figure 2a). Both pre- and post-SPION R2* liver maps were used in the first approach (*_PRE-POST_R2****), but only a single post-SPION R2* liver map was used in the second approach (*_ONLY-POST_R2**).A threshold (*_THRES-MEAN_R2**) was calculated using the middle value of *_TUMOR-MEAN_R2** (an average of all voxel values in *_TUMOR_R2**) and *_LIVER-MEAN_R2** (an average of all voxel values in *_LIVER_R2**). All voxel values in *_TUMOR_R2** and *_LIVER_R2** were totaled and the sum was divided by the number of voxels in the tumor and liver contours, respectively (see Figure 2b).A voxel-wise FLPV was automatically determined by comparing each voxel value of an R2* liver map to the *_THRES-MEAN_R2** (see Figure 2c). In the *_PRE-POST_R2** approach, if each voxel was greater than *_THRES-MEAN_R2**, it became a voxel of FLPV. However, the opposite worked in the *_ONLY-POST_R2** approach. The voxel-wise FLPV was saved as a new contour together with the physician-identified tumor and liver contours and transferred to MiM. This study developed the in-house auto-contouring tool in Matlab version 9.10 (The MathWorks, Natick, MA, USA). A long TE (13.8 ms) negatively enhanced more R2* in the liver map and led to an improvement in heterogeneity in the R2* liver map, so this study utilized the R2* liver map of a long TE (13.8 ms) for auto-contouring FLPV.

We evaluated the impact of SPION on T2*- and R2*-MRI by examining 2D and 3D multi-echo image sets from pre- to post-SPION. We visually inspected T2*-images and R2* changes in tumor and FLP regions across 16 echo 2D image sets and 3 echo 3D image sets. In addition, the heterogeneity of the R2* liver map was characterized by quantifying two voxel-wise FLPVs using the *_PRE-POST_R2** and the *_ONLY-POST_R2** approaches in an auto-contouring tool. In MiM, both voxel-wise FLPVs were overlaid on the liver image to evaluate the efficiency of our auto-contouring tool for the 12 patients with liver cancers. The tumor contour of each patient was visually inspected to find the area of overlap between tumor and FLPV contours as a function of effectiveness in the presence of the SPION enhancement. Quantitative statistical comparison between pre- and post-R2* changes was determined using an average of R2* in tumor and liver contours in a paired Student’s *t*-test.

## 3. Results

All patients successfully completed both the pre- and post-SPION MRI sessions without any SPION-related adverse events.

### 3.1. Patients

Table 1 presents the demographics of all patients enrolled in this study. Eight male and four female patients with primary (*n* = 8) and metastatic (*n* = 4) liver cancers enrolled, with a mean liver volume of 1594 ± 519 mL. We developed and optimized MRI sequences with Ex-BH while we scanned the first five patients, and we used the same MRI sequences for all twelve patients in this study. An Ex-BH of up to 24 s was a challenge for all liver cancer patients, but most of the 12 liver cancer patients performed well, except for P01 in the first MRI session and P12 in both MRI sessions. Patients performed their Ex-BH better in the second MRI session compared to the first MRI session.

### 3.2. Multi-Echo 2D/3D T2*- and R2*-MRI

Multi-echo 2D T2*-MR image sets were acquired for four liver cancer patients before and after the SPION injection. Figure 3 shows 2D image sets of multi-echo 2D T2*-MRI acquired in pre- (Figure 3a) and post-SPION (Figure 3b) MR simulations. The T2* of all 2D image sets continuously decreased across 16 echoes for both pre- and post-SPION 2D T2*-MRI, but it changed more on 2D image sets of post-SPION T2*-MRI. In addition, the R2* continuously decreased between pre- (Figure 3c) and post-SPION regions (Figure 3d), with it being calculated within two circles with a 2 cm diameter placed on tumor (red circle) and FLP (blue circle)). The R2* changed more in the post-SPION 2D R2*-MRI.

The maximum difference in R2*s in liver and tumor regions between pre- and post-SPION were found around TE = 13.8 ms, as shown in the dotted box (see Figure 3c,d). The R2* change in the liver region was more noticeable (*p*-value < 0.001 in all four patients) than the R2* change in terms of the tumor in between pre- and post-SPION (Figure 3d).

Multi-echo 3D T2*-MR image sets were acquired in 12 liver cancer patients before and after the SPION injection. Figure 4 shows a multi-echo 3D T2*-MRI of pre- and post-SPION (Figure 4a,b). Two contours of tumor (red) and liver (cyan) identified by a physician were used to perceive their R2* changes. The R2* change from pre- to post-SPION in the tumor contour was shown across three echoes (Figure 4c), and the R2* change in terms of pre- and post-SPION in the liver contour was also shown across three echoes (Figure 4d).

As was observed in the 2D scans, both 3D R2*(s) in terms of the tumor and liver continuously decreased across three echoes, but the R2* of the liver decreased more than the R2* of the tumor. The maximum difference in the R2* in the tumor and liver between pre- and post-SPION was found at a TE of approximately 13.8 ms (see Figure 3c,d). The R2* change in the liver (4.56 Hz) was three times greater than the R2* change in the tumor (1.47 Hz) in between pre- and post-SPION (see Figure 4d).

### 3.3. Characterizing Liver Heterogeneity in R2*

The heterogeneity of R2* in the liver increased in accordance with the number of KC(s), which reflects healthy hepatic liver parenchyma [28,29]. Figure 5 shows the change in R2* measured in patient P02 before and 72 h after an IV SPION injection. The three arrows point to the tumor where there was no difference, i.e., the bright and dark areas in Figure 5a, Figure 5b and Figure 5c, respectively. There was no difference between the pre-SPION R2* in the FLP and the pre-SPION R2* in the tumor, (Figure 5a), but the darker background in the FLP indicated that the post-SPION R2* had more heterogeneity to identify the tumor (see Figure 5b). The R2* change (*_PRE-POST_R2**) was minimal in the tumor but considerable in the FLP (Figure 5c).

The R2* heterogeneity of pre-SPION in the liver was insufficient to differentiate between the tumor and healthy FLP in Figure 5a; however, the difference was clearly identifiable in Figure 5b due to the impact of SPION, which negatively enhanced the R2* in the FLP but very minimally enhanced the R2* in the tumor. Figure 5c illustrates the minimal R2* change in the tumor (an average R2* change in terms of the tumor = 0.06 Hz), where it is dark, (i.e., see the black arrow pointing the tumor), and the noticeable change of R2* in FLP (an average R2* change in terms of the FLV = 3.72 Hz), where it is bright. Compared to pre-SPION, the liver heterogeneity of R2* in post-SPION considerably improved in nine liver cancer patients, slightly improved in P11 and P12, and was minimally altered in P08.

Overall, SPIONs negatively enhanced the R2* of healthy livers and tumors across all patients examined. Figure 6 shows the overall R2* change in livers and tumors before and after the SPION enhancement as averaged between 12 patients with liver cancers.

The R2* of both healthy livers and tumors decreased after the negative SPION enhancement, but the R2* of the liver decreased more than two-fold compared to the R2* of the tumor (Student’s *t*-test *p*-value = 0.011) in the 12 liver cancer patients. The R2* of healthy liver reduced 2.0 Hz (or more) in nine liver cancer patients, except for patients P08, P11 and P12. In these three liver cancer patients, the R2* of healthy liver decreased by less than 0.6 Hz. In contrast, the R2* of tumor with a change greater than 2.0 Hz was found in two liver cancer patients (P06 and P10).

### 3.4. Characterizing FLP Using an in-House Tool

Figure 7 shows the FLPV identified in the auto-contouring tool with the *_THRES-MEAN_R2** of the *_PRE-POST_R2** approach. The FLPV was clearly identified with minimal overlapping for three tumors in patient P02 (Figure 7a) and a tumor in P09 (Figure 7b). The FLPVs in patients P02 and P09 were 1186.0 mL (65.5%) out of 1811.3 mL and 357.6 mL (56.1%) out of 637.6 mL, respectively. P02 had three tumors, contoured in red, (Tumor 1), green, (Tumor 2) and blue, (Tumor 3). P09 had a single tumor, contoured in red. All tumor illustrations demonstrate the efficiency of auto-contouring the FLPV on axial, sagittal and coronal 3D planes.

This study also utilized the *_ONLY-POST_R2** approach to characterize the FLPV with our auto-contouring tool. Figure 8 shows the FLPV identified in the auto-contouring tool with *_THRES-MEAN_R2** in the *_ONLY-POST_R2** approach. The FLPV was clearly identified with minimal overlapping for three tumors in patient P02 (Figure 8a) and a tumor in P09 (Figure 8b). The FLPVs in P02 and P09 were 1399.3 mL (77.3%) out of 1811.3 mL and 494.0 mL (77.5%) out of 637.6 mL, respectively. The three tumors in P02 and the single tumor in P09 demonstrate the efficiency of auto-contouring the FLPV on axial, sagittal and coronal 3D planes.

Table 2 shows the tumor and healthy liver volumes delineated by a physician as well as the FLPV(s) identified in both *_PRE-POST_R2** and *_ONLY-POST_R2** approaches across the 12 liver cancer patients.

Among the 12 patients, the volume of the tumor and liver randomly varied (*R^2^* = 0.41) between 1.8 mL and 227.6 mL and 637.6 mL and 2637.8 mL, respectively. An average of *_PRE-POST_ FLPV* (%) and *_ONLY-POST_ FLPV* (%) was calculated as 60.2 ± 12.8% and 63.1 ± 9.9%, respectively. The largest and smallest differences in terms of FLPV between *_PRE-POST_ FLPV* (%) and *_ONLY-POST_ FLPV* (%) were found in P05 (25.3%) and P10 (2.7%), respectively. A difference of less than 15% between *_PRE-POST_ FLPV* (%) and *_ONLY-POST_ FLPV* (%) was found in nine patients; three patients had greater differences (P05 = 25.3%, P09 = 21.4% and P12 = 21.5%). The correlation between *_PRE-POST_ FLPV* (%) and *_ONLY-POST_ FLPV* (%) was not significant (*R^2^* = 0.65). This study was limited to the optimization of the *_THRES-MEAN_R2** for *_PRE-POST_R2** and *_ONLY-POST_R2** approaches instead of the simple use of the *_THRES-MEAN_R2**. The FLPV(s) of *_PRE-POST_R2** and *_ONLY-POST_R2** approaches identified in this study had no significant correlation to the volume of tumor (*R^2^* = 0.47 for the *_PRE-POST_R2** approach and *R^2^* = 0.43 for the *_ONLY-POST_R2** approach) but had considerable correlation to the volume of liver (*R^2^* = 0.71 for *_PRE-POST_R2** and *R^2^* = 0.85 for *_ONLY-POST_R2**).

## 4. Discussion

Liver radiation treatment planning often requires that the FLPV be measured in each patient to determine the radiation dose limit to residual hepatic parenchyma in patients with pre-existing hepatic conditions (hepatic cirrhosis, prior liver-directed therapies and hepatic resection). In this study, we utilized a SPION-CA for each patient to improve liver heterogeneity in T2*-MRI and calculated the voxel-wise R2* relaxation rates for the auto-contouring of the FLPV. Using the FLPV, we demonstrated the efficiency of our auto-contouring tool to identify the FLPV while overlaying the FLPV on a liver image to visually inspect the tumor and the FLPV. The FLPV, when accurately identified in each patient, can be applied for the avoidance of radiation-induced liver damage.

The SPIONs, at 72 h after the IV injection, clearly decreased the R2* of the liver, which was almost double the R2* of the tumor and proved essential for the auto-contouring tool. We found that approximately 60% of the total liver volume in the range [39.8%, 77.5%] was a healthy FLPV in our auto-contouring tool, and this technique helps to ensure that the FLPV is not exposed to high dose radiation during liver SBRT [11,12,13]. It may also guide hepatectomy in sparing the FLPV while ensuring the surgical resection margin [30,31,32].

The FLPV in each patient was identified by comparing the R2* values in each voxel to a threshold determined through the two approaches proposed in this study. In the first approach, both pre- and post-SPION R2*(s) were used to determine a threshold for our auto-contouring tool, so it required two MRI sessions. Meanwhile, the second approach required a single MRI session because it uses a single post-SPION R2*. In addition, the second approach was much simpler because it did not need image registration between pre- and post-SPION R2*(s) and therefore eliminated image registration uncertainty. The second approach could be further improved with the use of other thresholds with other methodologies [33,34].

The SPION-CA Ferumoxytol^®^ negatively enhanced the R2* of the liver in 9 out of 12 liver cancer patients (see Figure 6), with a considerable change; however, there was a minimal change in P11 and P12. Interestingly, the SPION-CA positively enhanced the R2* of tumors in P10 during the post-SPION T2*-MRI because of a truncated liver, which led to zero voxel values in the truncated liver and resulted in a miscalculated R2* for P10. The size of P10’s liver was especially large, so a pre-determined FOV could not cover the entire liver, requiring us to adjust the FOV for each patient. To enhance the R2* of the livers in P11 and P12, a higher dose of SPION-CA (i.e., up to 7.5 mg/kg for prostate [29,35], 3–7 mg/kg for brain [36] and 3–4 mg/kg for vessel [37]) was needed to completely darken the FLPV and thus relatively brighten non-FLPV in the post-SPION R2*-MRI.

The FLPV, defined by KC clusters on SPION-enhanced MRI, caused considerable R2* changes between pre-and post-SPION and within post-SPION. Non-FLPV related to liver tumors, hepatic vessels, liver fibrosis and scarring associated with hepatic cirrhosis or prior liver irradiation and resulted in KC loss and no (or minimal) R2* change [7]. For SPION enhancement, other SPION-CA(s) such as Ferumoxtran-10 (Combidex; Advanced Magnetics; Sinerem, Guerbet) [38] and Ferucarbotran (Resovist, Bayer Healthcare) [39] could be used as an alternative of Ferumoxytol^®^ to produce the same (or similar) outcome to improve liver heterogeneity for distinguishing between the FLPV and non-FLPV in R2*, which we demonstrated in Figure 7 and Figure 8.

Holding their breath for up to 24 s was challenging for all the liver cancer patients, but it was essential to eliminate motion artifacts during multi-echo 3D T2*-MRI. Therefore, we asked all the liver cancer patients to slowly breathe in when they could not hold their breath for an extended time. A couple of the liver cancer patients could not fully comply, indicating the need for training sessions for breath holds and the use of visual guidance for better image quality without motion artifacts [40]. In addition, the moderate to severe hearing loss in geriatric patients also increased the difficulty in relaying instructions since hearing aids are not permitted in MRI [41,42]. For these patients, the instructions were repeated as often as necessary.

A limitation of the present study was that the R2* changes were determined using only two MRI sessions: one before and the other after the SPION injection. The accuracy of the FLPV measured in our auto-contouring tool was only qualitatively evaluated while visually inspecting the overlapping contour between the FLPV and non-FLPV (tumor, hepatic vessels, liver fibrosis and scarring associated with hepatic cirrhosis). In future studies, multiple T2*-MRIs should be tested, acquiring more T2*-MRIs after radiotherapy and/or liver resection, and the uncertainty in terms of image registration should be improved by using deformable image registration [43]. These studies are ongoing at our institution and include the quantitative analysis of the FLPV to assess the accuracy of auto-contouring by comparing it with the FLPV measured by other imaging techniques for cross validation (i.e., SPECT and MRI) [44,45].

Building on our previous studies that used the nuclear medicine platform for liver SBRT planning [22,23,24,25,26], this study demonstrated that the use of a SPION-CA can enhance voxel-wise liver heterogeneity, and it can be advanced for the quantitative auto-contouring of FLPV(s) for the functional planning of liver SBRT. The present study used a SPION-CA to identify the FLPV, but the same approach could be used for visualizing liver tumors and the FLPV during MR-guided online adaptive radiotherapy [39]. In addition, T2*- and R2*-MRI with free-breathing [46] can assist in quantifying the FLPV without respiratory-induced motion artifacts.

## 5. Conclusions

This was the first study to directly evaluate the impact of SPIONs on functioning liver parenchyma with T2*- and R2*-MRI in patients with primary and metastatic liver cancers with an Elekta Unity MR-Linac. By using SPIONs, liver heterogeneity was improved across pre- and post-SPION MRI sessions, which were carried out 72 h apart. Our auto-contouring tool identified an average of 60% of the liver (39.8–77.5%) as the FLPV and performed well in 10 out of 12 liver cancer patients. These results demonstrate that our auto-contouring tool with SPIONs can facilitate the heterogeneous R2* of the liver, which is a desirable technique for achieving accurate liver SBRT planning.

## Figures and Tables

**Figure 1 cancers-14-05269-f001:**
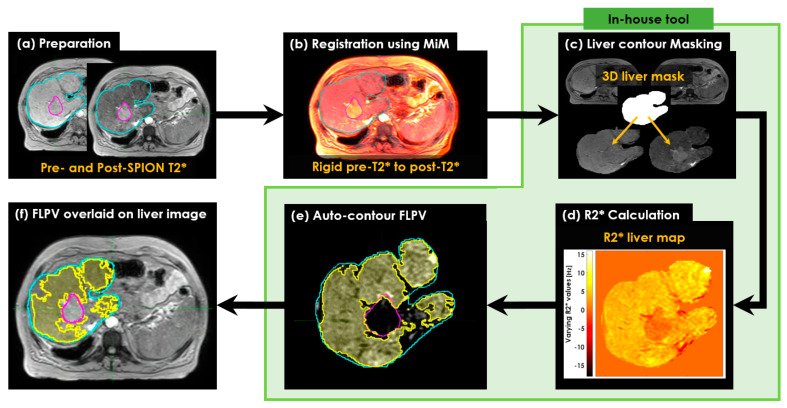
The workflow of characterizing FLP and non-FLP in the R2* liver map. (**a**) Preparation of pre- and post-SPION T2* image sets; (**b**) registration of the two pre- and post-SPION T2* image sets to copy the liver (Cyan color) and tumor (Magenta color) contours manually identified by a physician; (**c**) masking the liver contour to extract the corresponding liver volume in voxels; (**d**) calculation of R2* relaxation rates; (**e**) auto-contouring FLPV colored area in yellow; and (**f**) overlaid FLPV on liver image to evaluate the efficiency of an auto-contouring tool for further sparing of healthy tissues during radiotherapy and liver resection.

**Figure 2 cancers-14-05269-f002:**
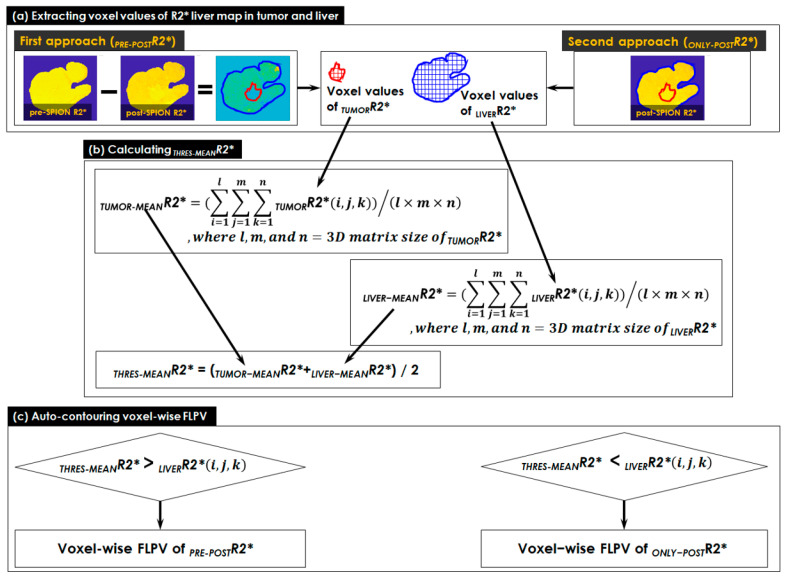
The two approaches for auto-contouring voxel-wise FLPV. (**a**) Extracting voxel values of R2* liver map in tumor and liver contours, which used both pre- and post-SPION R2* liver maps in the first approach (*_PRE-POST_R2**) and only a single post-SPION R2* liver map in the second approach (*_ONLY-POST_R2**); (**b**) calculating a threshold (*_THRES-MEAN_R2**) using the middle value of *_TUMOR-MEAN_R2** (an average of all voxel values in *_TUMOR_R2**) and *_LIVER-MEAN_R2** (an average of all voxel values in _LIVER_*R2**); and (**c**) auto-contouring voxel-wise FLPV determined by comparing each voxel value of R2* liver map to *_THRES-MEAN_R2**. If each voxel was greater than *_THRES-MEAN_R2**, it became a voxel of FLPV in the *_PRE-POST_R2** approach, but the opposite worked in the *_ONLY-POST_R2** approach. FLPV was saved as a new contour together with the physician-identified tumor and liver contours and transferred to MiM for further overlaying on the liver image.

**Figure 3 cancers-14-05269-f003:**
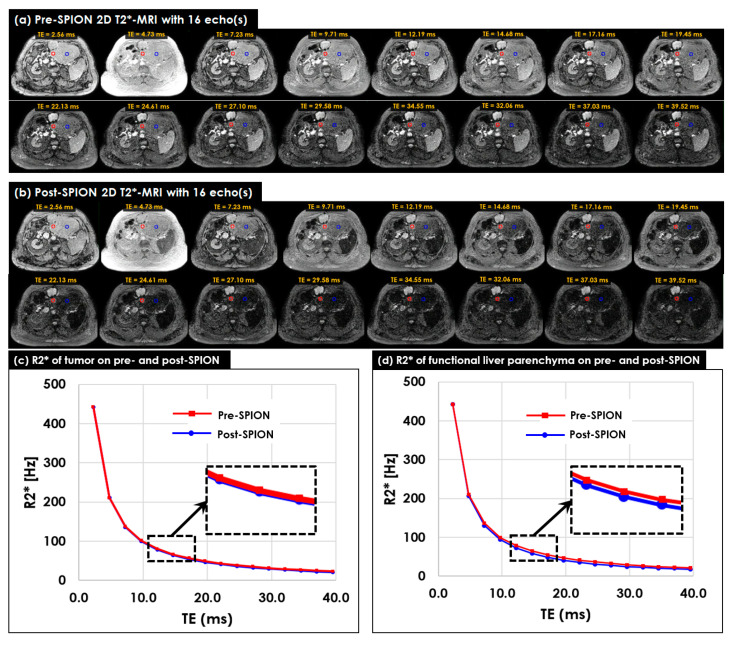
Pre- and post-SPION multi-echo 2D T2*-MRI and R2*-MRI in P09. (**a**) Pre- and (**b**) post- SPION 2D T2*-MRI with 16 echoes acquired at every 2.3 ms before and after 72 h after the SPION injection. Two circles with a 2 cm diameter were placed on the regions of tumor (red circle) and FLP (blue circle) to measure the changes in terms of R2* within the two circles at every echo on (**c**) pre- and (**d**) post-SPION R2*-MRI. An average of the R2* in the regions of tumor and FLP was used to depict a varying R2* [Hz] across 16 echoes. The R2* in the FLP region, compared to the R2* in the tumor region, noticeably changed (Student’s *t*-test *p*-value < 0.001), as shown in the dotted rectangles in (**c**,**d**) at around TE = 13.8 ms.

**Figure 4 cancers-14-05269-f004:**
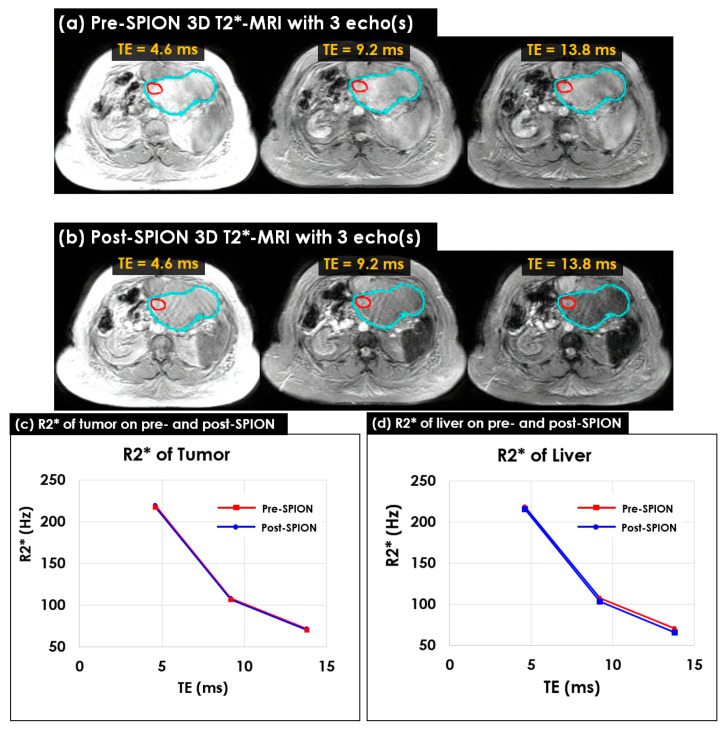
Pre- and post-SPION multi-echo 3D T2*-MRI and R2*-MRI in P09. (**a**) Pre- and (**b**) post- SPION 3D T2*-MRI with three echoes acquired at every 4.6 ms before and after 72 h after the SPION injection; (**c**) the R2* change in tumor between pre- (red line) and post-SPION (blue line) across three echoes, and (**d**) the R2* change in liver between pre- (red line) and post-SPION (blue line) at three echoes. An average of R2* in the two contours of tumor (red) and liver (cyan) were used to depict varying R2*(s). The R2* of the liver noticeably changed more than the R2* of the tumor, especially at TE = 13.8 ms.

**Figure 5 cancers-14-05269-f005:**
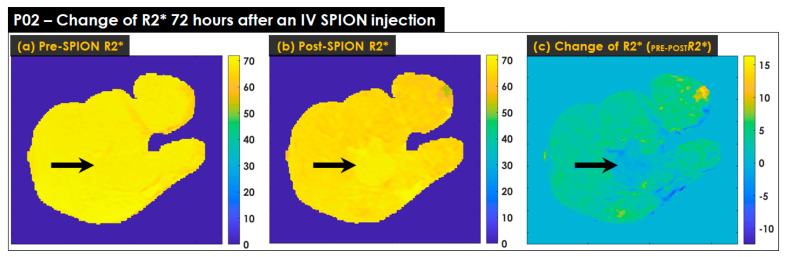
The change in R2* measured in patient P02 before and 72 h after an IV SPION injection. (**a**) The R2* (TE = 13.8 ms) of pre-SPION; (**b**) the R2* (TE = 13.8 ms) of post-SPION, and (**c**) the change in R2* calculated by subtracting (**a**) to (**b**). The black arrows point to the tumor in each panel. In (**a**), pre-SPION R2*, there is no visual difference between tumor and liver tissue. The tumor is visually lighter in (**b**), post-SPION R2*, and darker in (**c**) compared to surrounding tissues. The change in R2* was minimal on the tumor but considerable for FLP in *_PRE-POST_R2**, as shown by the coloration of FLP but not the tumor in (**c**).

**Figure 6 cancers-14-05269-f006:**
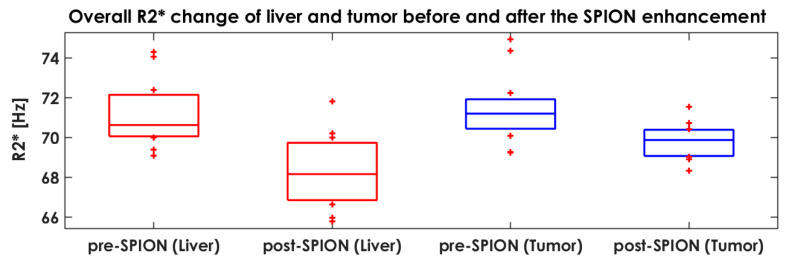
Overall R2* change in healthy livers and tumors before and after the SPION enhancement. The first two (red color) boxplots represent the R2* of the liver in pre- and post-SPION, and the last two (blue color) boxplots represent the R2* of the tumor in pre- and post-SPION.

**Figure 7 cancers-14-05269-f007:**
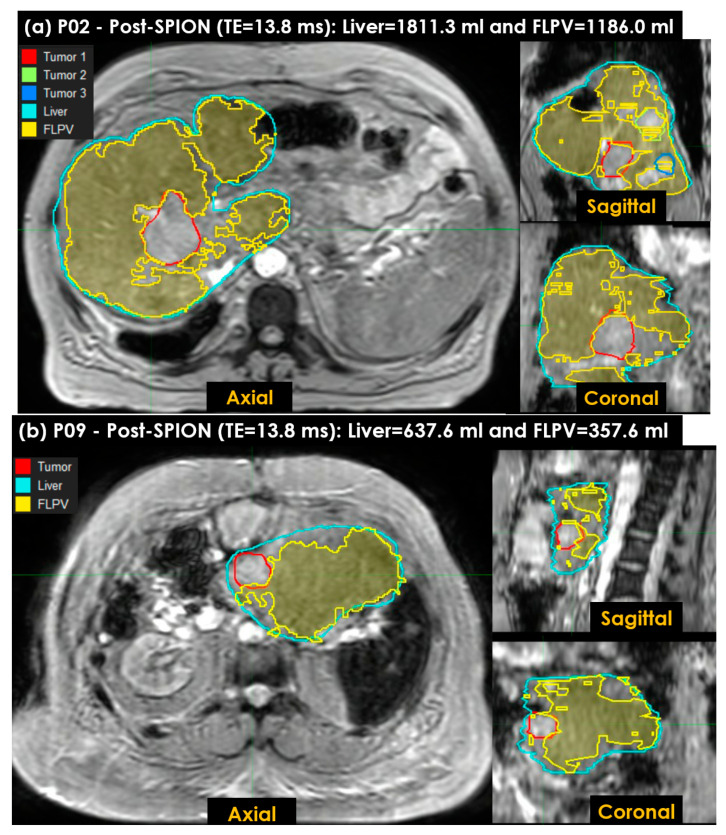
The functional liver parenchyma volume(s) FLPV(s), identified using the *_THRES-MEAN_**R2** of the first *_PRE-POST_R2** approach in an auto-contouring in-house tool, were overlaid on liver images. (**a**) Three tumors in patient P02, contoured in red (first tumor), green (second tumor) and blue (third tumor), are illustrated to demonstrate the efficiency of the in-house tool for identifying the FLPV; (**b**) a single tumor in patient P09, contoured in red, is also illustrated. Both FLPVs of P02 and P09 are clearly distinguished from all tumors with minimal overlapping between the FLPV and non-FLPV (tumor, hepatic vessels, liver fibrosis and scarring associated with hepatic cirrhosis).

**Figure 8 cancers-14-05269-f008:**
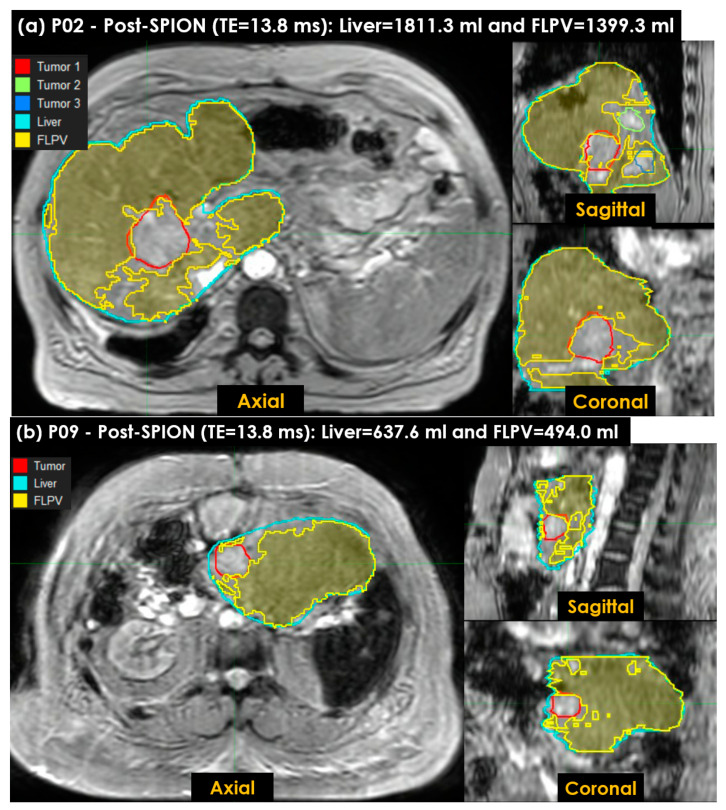
The functional liver parenchyma volume(s) FLPV(s), identified using the *_THRES-MEAN_**R2** of the second *_ONLY-POST_R2** approach in an auto-contouring in-house tool, were overlaid on liver images. (**a**) Three tumors in patient P02, contoured in red (first tumor), green (second tumor) and blue (third tumor), are illustrated to demonstrate the efficiency of the in-house tool for identifying the FLPV; (**b**) a single tumor in patient P09, contoured in red, is also illustrated. Similarly, both the FLPVs of P02 and P09 are clearly distinguished from all tumors with minimal overlap between the FLPVs and non-FLPVs (tumor, hepatic vessels, liver fibrosis and scarring associated with hepatic cirrhosis).

**Table 1 cancers-14-05269-t001:** Demographic and disease profile of liver cancer patients.

Patient #	Gender	Age	Child–Pugh Score	Diagnosis	Tumor Location	Liver Volume (mL)
P01	M	80	-	HCC	Liver Seg 4	1635.6
P02	M	55	-	Metastases	Liver Right lobe	1811.3
P03	F	79	Child–Pugh B Nash Cirrhosis	HCC	Liver Seg 6 and Seg 2	1439.4
P04	M	53	-	Metastases	Liver Seg 8	1267.2
P05	M	62	Child–Pugh A Nash Cirrhosis	HCC	Liver Seg 7	1791.8
P06	M	53	-	Metastases	Liver Seg 8	1858.5
P07	F	81	Child–Pugh B Cirrhosis	HCC	Liver Seg 7	901.1
P08	M	60	Child–Pugh B Cirrhosis	HCC	Liver Seg 4A/8	1985.7
P09	F	56	-	Metastases	Right Hepatic Lobe	637.6
P10	M	74	Child–Pugh A Cirrhosis Nash	HCC	Liver Seg 6 and Seg 2	2637.8
P11	M	73	Child–Pugh A Cirrhosis Nash	HCC	Liver Seg 7 Hilum	1187.1
P12	F	86	Child–Pugh A Cirrhosis Nash	HCC	Liver Seg 5/6	1971.5

M = male; F = female; HCC = hepatocellular carcinoma; Seg = segment.

**Table 2 cancers-14-05269-t002:** Tumor and healthy liver volumes and identified FLPVs in both *_PRE-POST_**R2** and *_ONLY-POST_R2** approaches across 12 liver cancer patients. The percentage (%) of *_PRE-POST_ FLPV* and *_ONLY-POST_ FLPV* was calculated using *_PRE-POST_ FLPV* (%) = (*_PRE-POST_ FLPV* (mL)/liver volume (mL) × 100) and *_ONLY-POST_ FLPV* (%) = (*_ONLY-POST_ FLPV* (mL)/liver volume (mL) × 100).

Volume	Patients											
	P01	P02	P03	P04	P05	P06	P07	P08	P09	P10	P11	P12
Tumor (mL)	51.2	77.3	6.1	1.8	7.8	42.3	6.9	13.5	12.0	227.6	45.9	6.1
Liver (mL)	1635.6	1811.3	1439.4	1267.2	1791.8	1858.5	901.1	1985.7	637.6	2637.8	1187.1	1971.5
*_PRE-POST_ FLPV* (mL)	1256.3	1186.0	957.3	729.9	713.7	1289.7	734.5	1207.1	357.6	1681.0	505.0	830.4
*_PRE-POST_ FLPV* (%)	76.8	65.5	66.5	57.6	39.8	69.4	81.5	60.8	56.1	63.0	42.5	42.1
*_ONLY-POST_ FLPV* (mL)	1060.0	1399.3	766.7	590.0	1166.1	1212.0	691.8	1049.4	494.0	1610.1	636.6	1253.8
*_ONLY-POST_ FLPV* (%)	64.8	77.3	53.3	46.6	65.1	65.2	76.8	52.8	77.5	61.0	53.6	63.6

## Data Availability

The data can be shared up on request.

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
