# Peer review of "Quantifying Liver Heterogeneity via R2*-MRI with Super-Paramagnetic Iron Oxide Nanoparticles (SPION) to Characterize Liver Function and Tumor"

_cancers, 2022, doi:10.3390/cancers14215269_

Round 1
Reviewer 1 Report
This work seeks to evaluate the impact of SPIONs on FLPV in R2*-MRI for liver cancer patients. This work also introduces a in-house software with an auto contouring tool that can automatically characterize FLP. Below are some specific comments:
1. The language of the manuscript has to be improved. Many awkward sentences required a few reads to understand
2. [first paragraph of discussion] while this work was able to demonstrate that the auto-contouring tool can yield great results, the claim that it can be applied to liver radiation treatment planning and hepatectomy lack substance. I suggest mentioning how the specific results obtained in this study can directly improve these applications.
3. [line 311-313] As the authors stated, SPION-CA somehow positively enhanced the R2* of tumors for p10. I think a more thorough explanation is needed this seems to contradict the other results.
4. Not many details are given about the auto-contouring tool. Only results are presented, but no detail was given on how this tool works and how the tool was able to obtain the results shown in the paper.
5. At a glance of the results, the auto contouring tool was able to identify the tumor and FLV using R2*. However, I don’t see any validation work. For example, you can consider assessing the accuracy of auto contouring by comparing it with delineation by a physician.
Author Response
Thank you for the review and comments.
I do appreciate if you find an attached file for a point-by-point response.

Reviewer 2 Report
In current research article author choose interesting study about SPIONs enhanced liver heterogeneity in the R2* of liver which can be utilized to characterize liver function and tumors by developing an auto-contouring tool to determine functional liver parenchyma volume (FLPV). The author states that the liver is overlaid with FLPV to evaluate the efficiency of the auto-contouring tool. Although research area is novel and upto the mark but following minor correction is needed before consideration of publication.
1. Grammatical mistakes and sentence rephrasing required throughout the article.
2. Introduction and discussion part may be improved.
3. Schematic/flow chart of overall work is required.
4. Some more relevant discussion is required, plenty of work has been done but specific and appropriate pattern should be followed o that articles catch the reader’s attention.
After successful modification and minor correction with refinement of articles, work may be consider for publication.
Author Response
Thank you for the review and comments.
I do appreciate if you find the attached file for a point-by-point response.

Round 2
Reviewer 1 Report
The manuscript has been significantly improved. The details added greatly enhanced the presentation of the study. The response letter addressed all of the major concerns I had. From a technical perspective, I think the manuscript is fine. A lot of the language issues from the first version were also fixed; however, I am still seeing grammar mistakes and awkward sentences.
Author Response
Thank you for the review of our manuscript and this opportunity to improve the manuscript.
The original queries and requests are reproduced verbatim in black. Our responses to these comments are written in blue.
Reviewers' comments:
Reviewer #1.
Thank you for the review and comments.
The manuscript has been significantly improved. The details added greatly enhanced the presentation of the study. The response letter addressed all of the major concerns I had. From a technical perspective, I think the manuscript is fine. A lot of the language issues from the first version were also fixed; however, I am still seeing grammar mistakes and awkward sentences.
We apologize for the gramma mistakes and awkward sentences in the revised manuscript. A senior publication support editor has revised it to improve all grammatical mistakes and awkward sentences for smooth and comfortable reading. All authors have confirmed the current manuscript.
